# Antifungal Activity of Mexican Propolis on Clinical Isolates of *Candida* Species

**DOI:** 10.3390/molecules27175651

**Published:** 2022-09-01

**Authors:** Claudia Rebeca Rivera-Yañez, Porfirio Alonso Ruiz-Hurtado, Julia Reyes-Reali, María Isabel Mendoza-Ramos, María Elena Vargas-Díaz, Karla Mariela Hernández-Sánchez, Glustein Pozo-Molina, Claudia Fabiola Méndez-Catalá, Gina Stella García-Romo, Alexander Pedroza-González, Adolfo René Méndez-Cruz, Oscar Nieto-Yañez, Nelly Rivera-Yañez

**Affiliations:** 1Carrera de Médico Cirujano, Facultad de Estudios Superiores Iztacala, Universidad Nacional Autónoma de Mexico, Tlalnepantla 54090, Mexico; 2Unidad de Morfofisiología y Función, Laboratorio de Inmunología, Facultad de Estudios Superiores Iztacala, Universidad Nacional Autónoma de Mexico, Tlalnepantla 54090, Mexico; 3Laboratorio de Toxicología de Productos Naturales, Departamento de Farmacia, Instituto Politécnico Nacional, Escuela Nacional de Ciencias Biológicas, Av. Wilfrido Massieu, Gustavo A. Madero 07738, Mexico; 4Laboratorio de Química de Productos Naturales, Departamento de Química Orgánica, Escuela Nacional de Ciencias Biológicas, Instituto Politécnico Nacional, Prol. de Carpio y Plan de Ayala, Ciudad de México 11340, Mexico; 5Laboratorio de Genética y Oncología Molecular, Laboratorio 5, Edificio A4, Facultad de Estudios Superiores Iztacala, Universidad Nacional Autónoma de Mexico, Tlalnepantla 54090, Mexico; 6División de Investigación y Posgrado, Facultad de Estudios Superiores Iztacala, Universidad Nacional Autónoma de Mexico, Tlalnepantla 54090, Mexico

**Keywords:** Mexican propolis, anti-*Candida* activity, germ tube, bioactive compounds, chemical composition

## Abstract

Infections caused by micro-organisms of the genus *Candida* are becoming a growing health problem worldwide. These fungi are opportunistic commensals that can produce infections—clinically known as candidiasis—in immunocompromised individuals. The indiscriminate use of different anti-fungal treatments has triggered the resistance of *Candida* species to currently used therapies. In this sense, propolis has been shown to have potent antimicrobial properties and thus can be used as an approach for the inhibition of *Candida* species. Therefore, this work aims to evaluate the anti-*Candida* effects of a propolis extract obtained from the north of Mexico on clinical isolates of *Candida* species. *Candida* species were specifically identified from oral lesions, and both the qualitative and quantitative anti-*Candida* effects of the Mexican propolis were evaluated, as well as its inhibitory effect on *C. albicans* isolate’s germ tube growth and chemical composition. Three *Candida* species were identified, and our results indicated that the inhibition halos of the propolis ranged from 7.6 to 21.43 mm, while that of the MFC and FC_50_ ranged from 0.312 to 1.25 and 0.014 to 0.244 mg/mL, respectively. Moreover, the propolis was found to inhibit germ tube formation (IC_50_ ranging from 0.030 to 1.291 mg/mL). Chemical composition analysis indicated the presence of flavonoids, including pinocembrin, baicalein, pinobanksin chalcone, rhamnetin, and biochanin A, in the Mexican propolis extract. In summary, our work shows that Mexican propolis presents significant anti-*Candida* effects related to its chemical composition, and also inhibits germ tube growth. Other *Candida* species virulence factors should be investigated in future research in order to determine the mechanisms associated with antifungal effects against them.

## 1. Introduction

Throughout history, human beings have used natural products such as propolis to alleviate diseases of different etiologies [1]. In its extensive period of use, propolis has been used for the treatment of bacterial [2,3], fungal [4,5], viral [6,7], and parasitic infections [8,9]. At present, it is well known that it possesses anti-inflammatory [10,11], antitumor [12,13], antidiabetic [14,15], and immunomodulatory properties [16,17]; however, it is its antioxidant effect that stands out and is maintained in most types of propolis from different countries [18,19,20]. Its great diversity of biological properties can be attributed to its complex chemical composition, in which more than 600 different chemical compounds have been identified [16], derived from the wide range of raw materials that bees use to make this natural resinous product [21]. Although the components of propolis vary greatly depending on the geographical region, most studies have reported a large number of secondary metabolites, such as terpenoids and polyphenols (flavonoids, phenolic acids, and their esters), among others [16].

Numerous references can be found in the literature on the use of propolis to alleviate different conditions of microbial origin, such as the case of candidiasis—a disease caused by various species of the genus *Candida*. As these pathogens can cause infections of the skin, mucous membranes of the oral cavity, and genitourinary and gastrointestinal tracts [22,23,24,25,26,27,28,29], these yeasts are called opportunistic, because mildly immunocompromised individuals can frequently have recurrent infections of their oral cavity, called oral candidiasis; one of the most common species is the yeast *Candida albicans*, which is a member of the commensal microbial community in humans [22,23,30]. In addition, not only can this species cause infections, but non-albicans infections caused by *Candida* sp. may occur, which have presented an increase of up to 60% in all episodes of invasive candidiasis in some health centers [31,32]. Worldwide, the incidence of invasive *Candida* infections has been increasing at a rate of 700,000 cases per year, associated with considerable mortality [33,34].

The success of these micro-organisms is due to the different virulence factors that contribute to fungal pathogenesis; the most studied and significant characteristic is the morphological transition from yeast to hyphae, which facilitates the rapid response to different environments, as well as during host infection [35]. To invade and infect various body niches, such as the oral cavity, *Candida* sp. ditches with various virulence factors and pathogenicity mechanisms, such as the morphological transition between yeast and hyphal forms, the expression of adhesins and invasins on the cell surface, thigmotropism, the formation of biofilms, phenotypic switching, and the secretion of hydrolytic enzymes, which are considered to be virulence factors [36,37]. Together with their great capacity for adapting to fluctuations in environmental pH, metabolic flexibility, powerful nutrient acquisition systems, and robust stress response machinery, this makes the *Candida* genus a set of pathogens with great infectious potential [36,38].

Current treatments are aimed at specifically attacking the membrane, cell wall, and the metabolic pathways that synthesize their main components. This has resulted in the manufacture and use of five different classes of allopathic antifungals: polyenes, azoles, allylamines, echinocandins, and pyrimidines [39,40]. In the specific case of candidemias, the main treatments involve the use of polyenes, azoles, and echinocandins [41]. Polyenes bind specifically to ergosterol, the main component of the fungal cell membrane, creating pores and subsequent cell death [42,43,44,45]; the azole group inhibits lanosterol 14 α demethylase, a key enzyme in ergosterol biosynthesis, presenting a fungistatic effect [46,47,48,49]; and echinocandins act by inhibiting β-D-glucan synthase, an important enzyme in cell wall synthesis [50,51,52,53,54,55,56]. However, in practice, the inadequate treatment of *Candida* sp. infections can cause the fungus to spread from the surface of the body to the internal organs (kidneys, heart, brain) and blood, provoking life-threatening invasive infections [57]. This may be due to the irresponsible and abusive use of different antifungal treatments [58], which has triggered a great resistance of yeasts to conventional treatments such as fluconazole and amphotericin B [59], thus creating increasingly resistant strains.

The use of natural products is currently being discussed as an alternative to combat fungal infections [60]. A natural product recently studied worldwide is propolis, which has shown efficacy against different fungal strains, including *Candida* sp. [5,61,62,63,64,65,66,67,68,69,70,71,72,73]. In this sense, it has been reported that propolis samples from different countries such as Poland, Iran, Cameroon, Brazil, the Czech Republic, Ireland, and Germany present both qualitative and quantitative antifungal activity differently in reference *Candida* strains such as *C. albicans* (ATCC 10231, 90028, 66396; CBS 562; NR 29450; SC 5314), *Candida krusei* (ATCC 6258, 90878; CBS 573), *Candida parapsilosis* (ATCC 22019; CBS 604), and *Candida glabrata* (CBS 07; DSM 11226; LMA 90-1085), *Candida tropicalis* (ATCC 9968; CBS 94), and *Candida dubliniensis* (CBS 7987), and on clinical isolates of *Candida* obtained from smears of the mouth and throat, as well as fluid from the peritoneal cavity, bronchopulmonary lavage, stoma, blood, urine, feces, and anus, identified as *C. albicans*, *C. glabrata*, *C. krusei*, *C. dubliniensis*, *C. tropicalis*, and *C. parapsilosis* [61,63,64,66,68,69,70,73]. Considering the properties of propolis from different countries, in this work, we focus on evaluating the effect of Mexican propolis on clinical isolates of *Candida* sp. and its ability to inhibit germ tube formation, as well as determining its chemical composition.

## 2. Results

### 2.1. Identification of Clinical Isolates of Candida

Based on the morphology and coloration characterizing each *Candida* species, and following the manufacturer’s guidelines through CHROMagar^TM^ *Candida*, three samples were identified as *C. krusei,* corresponding to Clinical Case 1 (CC1), CC5, and CC9 (Figure 1A; lower half of Petri dish, rose colonies of *Candida*); six samples were identified as *C. albicans*, corresponding to CC2, CC3, CC4, CC7, CC8, and CC10 (Figure 1A, upper half of Petri dish; 1B, upper and lower half of Petri dish; and 1C, lower half of Petri dish; green colonies of *Candida*); and, finally, one sample was identified as *C. glabrata* (Figure 1C, upper half of Petri dish; cream colonies of *Candida*), corresponding to CC6. All *Candida* sp. were isolated from the tongue of the patients.

### 2.2. Evaluation of the Antifungal Activity of Propolis in Candida

We found that the Mexican propolis presented antifungal activity in the 10 different clinical isolates of *Candida* evaluated, but with distinct degrees of activity. The results are displayed in Table 1, indicating that *C. glabrata* from CC6 was the most sensitive to propolis, as it exhibited 21.43 ± 1.30 mm inhibition halos. In contrast, *C. krusei* of CC5 presented the smallest inhibition halos (only 7.60 ± 0.10 mm). In Figure 2, various representative photographs of the inhibition halos indicating the antifungal activity of the propolis are shown.

Interesting data were obtained from the quantitative analysis of the antifungal activity of Mexican propolis, as all *Candida* species presented different concentrations of inhibition. The CC2 and CC3 (both *C. albicans*) and CC6 isolates (corresponding to *C. glabrata*), presented an MFC of 312 µg/mL. In contrast, CC4, CC7, and CC10 (*C. albicans*) and CC5 (*C. krusei*) reported an MFC of 1250 µg/mL. In addition, the most sensitive sample to propolis was *C. albicans* from CC7, which exhibited an FC_75_ of 19 ± 0.0067 µg/mL and an FC_50_ of 14 ± 0.0031 µg/mL. In the same sense, CC4 and CC5 were the least susceptible samples to propolis, presenting an FC_75_ of 492 ± 0.0220 µg/mL and an FC_50_ of 244 ± 0.0080 µg/mL, respectively. The results of this test are detailed in Table 2.

### 2.3. Evaluation of Candida Germ Tube Growth Inhibition

The concentration at which germinative tube growth was completely inhibited in CC2 (Figure 3B) and CC8 was 1250 µg/mL. Similarly, in CC3, CC4, CC7 (Figure 3E), and CC10, the concentration of 2500 µg/mL completely inhibited the germ tube growth in these samples. Furthermore, when determining the concentration at which propolis inhibited 50% of germ tube growth in each of the clinical isolates, we determined the lowest IC_50_ (19 ± 0.0015 µg/mL) for CC10 and the highest (1291 ± 0.0141 µg/mL) for CC4; detailed data are provided in Table 3. In Figure 3, representative microphotographs (40×) of the inhibitory effect of propolis on the germ tube growth of *C. albicans* are shown.

### 2.4. Chemical Composition of Mexican Propolis

Methanol was used to optimize the extraction as, being a polar solvent, it facilitated the extraction of compounds such as flavonoid aglycones substituted with a large number of OH and methoxyl groups, as well as isoflavones, flavanones, flavones, and flavonols. Initial NMR analysis of the propolis extract evidenced and confirmed the presence of phenolic compounds, as the ^1^H NMR spectrum showed signals of aromatic protons at δ 6–8. Subsequently, a portion (1 g) of the extract was separated by column chromatography. From this fractionation, 189 eluates were obtained, which were monitored by TLC, and those fractions that showed an interesting composition were analyzed by ESI-MS and NMR. Based on this analysis, various compounds were identified (Figure 4). The ^1^H NMR spectrum of F40 (white amorphous solid) showed signals at δ 12, belonging to an OH bridged to a carbonyl, as well as at δ 7.41, indicating signals corresponding to a AA′BB′C system. Signals belonging to diasterotopic protons were observed at δ 3.23 and 2.77, and so, it was concluded that the compound present in this fraction was pinocembrin (1). The ^13^C spectrum showed, at δ 195.8, the signal of a carbon with a typical displacement of the carbonyl group, and the signals of carbons attached to oxygen at δ 166.6–162.57 were observed; in addition, ESI-MS [M-H]^−^ identified one main molecular ion displaying at *m*/*z* 255.0688, confirming the presence of this dihydroflavone [74]. In the same order, F75 (yellow oily liquid) was also analyzed, and its ^1^H NMR showed signals of isoflavones at approximately δ 8.07–6.21, which are characteristic of an AA′BB system and meta-coupled aromatic protons; in the high field, we found signs of methoxyl groups. ^1^H and ^13^C NMR data, along with the information previously reported, suggested the presence of the compound 5,7-dihydroxi-4′-methoxyisoflavone (Biochanin A) (2) [75]. For F80 (slightly yellow powder), ESI-MS (-) showed a peak at *m*/*z* 271.0622, similar to 5,6,7-trihydroxy-flavone, i.e., baicalein (3), and the ^1^H NMR signal exhibited the presence of hydroxyl groups on ring A. At the same time, the spectrum presented an AA′BB′C system at δ 7.72 and 7.55, indicating that ring B did not present any substituent group, shifts, or multiplicity in the signals, further indicating the presence of this compound [76,77]. On the other hand, the spectral data (^1^H and ^13^C NMR) of F101 (white powder/amorphous solid) showed the presence of another phenolic compound, which could be a derivative of pinobanksin—pinobanksin chalcone (4)—which was corroborated by ESI-MS (-), as we observed a main peak at *m*/*z* 271.0625, thus coinciding with previous reports [78]. Compound 5,8-dihydroxy-flavanone (yellow powder) (5) was identified in F131 by NMR. The ^1^H NMR spectrum showed important signals between 8.2 and 7.7 ppm, the AA′BB′C system of the ring C of the flavone was observed, and ortho-coupled aromatic protons were present at δ 6.70 and 6.40. In the ^1^H NMR spectrum obtained from fraction F161 (yellow powder), OH signals were observed at δ 12, while downfield signals of aromatic protons characteristic of phenolic compounds were also observed (7.5 to 6.7 ppm). At δ 6.33 and 6.20, two double signals were found belonging to the protons H-6 and H-8 on the aromatic ring A. In the ^13^C spectrum, we observed the signal of a carbonyl group at δ 187.6, the signals of carbons attached to oxygen at δ 161.4 to 154.2, and the characteristic signals of methoxyl carbons at approximately 72–64 ppm in the high field. This spectroscopic analysis, in addition to the ESI-MS [M-H]^−^ (*m*/*z* 315.0524) and comparisons with previous reports [79,80,81], allowed us to determine that rhamnetin (6) was present in this fraction. Spectral data can be found in the Appendix A.

## 3. Discussion

Natural products are an invaluable source of compounds with various medicinal properties. These bioactive molecules come from various sources, including terrestrial plants, micro-organisms, marine organisms, and terrestrial vertebrates and invertebrates. Microorganisms have provided great contributions such as penicillin and other antimicrobials. Similarly, compounds of a plant origin provide a host of new agents with therapeutic potential [82,83]. Propolis is a bee product elaborated by bees from compounds of a plant origin and has proven to be a great candidate for treating different health conditions [84].

Different studies worldwide have demonstrated that, among the biological properties of propolis in distinct regions of the world, antioxidant, anti-inflammatory, and antimicrobial effects are commonly observed with this natural product [14,60,85,86,87,88,89]. Nevertheless, these properties tend to vary according to the specific region in which the propolis samples are obtained, as the specific flora of the bees’ environment is different in each zone and continent of the world. Despite this, propolis tends to present some similar characteristics, which have been used for its classification into common classes [90,91,92]. Mexican propolis from the north region is usually classified as poplar-type propolis, as pinocembrin is characteristic in its chemical composition, which is considered as a marker of poplar-type propolis [89,91]. Accordingly, this same flavonoid was identified in our propolis extract, which was obtained from the north of Mexico. Therefore, both the chemical and biological properties of our extract may be similar to other poplar-type propolis, such as European and some South American propolis.

Among the different biological properties of propolis, its antifungal activity is re-markable. Some of the most interesting organisms which are directly related to infections of the oral cavity are the fungi of the genus *Candida*, which are opportunistic commensals that can produce infections in individuals with an immunosuppressed state caused by the intake of drugs, or through systemic diseases such as cancer or diabetes. Furthermore, the use of some dispositives in the oral cavity, such as prostheses infected with *Candida* sp. used in odontological treatments, can contribute not only to infection with this microscopical fungi, but also can have an influence on and promote the overgrowth of *Candida* sp., leading to the condition clinically known as candidiasis [93,94,95,96]. There has been an increasing interest in finding alternatives for the treatment of candidiasis, due to the increasing resistance of *Candida* sp. to classical drugs, without taking the side effects associated with them into account [72,94,97,98]. Natural products present an attractive option, due to their low or non-existent toxicity; furthermore, the rich chemical composition of propolis provides the advantage that some of the secondary metabolites present in it could act on some virulence factors that display *Candida* sp., contributing to the attenuation of the pathogenicity of these micro-organisms. This additional effect is aimed at inhibiting virulence rather than only the growth of fungus, which may impose weaker selective pressure on the development of drug resistance [99].

Considering the above, poplar-type propolis has presented anti-*Candida* properties, as is the case with European propolis, which, in comparison with two different samples of red and green Brazilian propolis, has been shown to have a higher microbicidal activity. Moreover, in that study, they tested the fungicidal effects of 50 propolis samples from Poland in five strains of *C. albicans* (ATCC 76615), and the mean MIC value of the propolis samples was lower than 25,000 µg/mL for all *Candida* strains [60]; in this sense, our poplar-type Mexican propolis sample presented higher fungicidal activity in comparison to European propolis, as the FC_50_ values were lower than 1000 µg/mL, in addition to the fact that the MFC values in both *C. albicans* and the other *Candida* strains tested in our work (i.e., *C. krusei* and *C. glabrata*) ranged between 1250 µg/mL and 312 µg/mL. It should be noted that the European propolis samples were tested on ATCC strains of *C. albicans*, while on the contrary, in our study, a Mexican propolis sample was tested in clinical isolates of *Candida*; therefore, the differences among the *Candida* strain samples are a factor that may have had an influence on the distinct susceptibility to the inhibitory effect of the propolis samples.

Moreover, our Mexican propolis sample also showed a better antifungal effect than Iranian propolis tested on *C. albicans* human isolates from oral cavity, vaginal, and nail infections. In this work, the authors reported that their Iranian propolis sample presented a mean MFC value of 1250 µg/mL and a mean MIC value of 360 µg/mL [88]; moreover, another work has reported MIC values of 2740 and 9010 µg/mL for Iranian alcoholic and hydroalcoholic propolis extracts in *C. albicans* isolates from the oral cavity of patients with colorectal cancer under chemotherapy [72]. Compared to our results, the mean MFC value of our Mexican propolis sample was 833 µg/mL, and the mean FC_50_ (65 µg/mL) value also was lower than that of the Iranian propolis. Interestingly, the range of MIC concentrations reported in *C. albicans* strains obtained from oral cavity injuries (120–970 µg/mL) was the same as that for *C. albicans* strains from vaginal infection and similar to that of the strains obtained from nail infection (120–480 µg/mL). In comparison, our propolis sample showed a range of values from 14 µg/mL to 237 µg/mL, as we only tested the propolis in samples obtained from the oral cavity. This could indicate that Mexican poplar-type propolis likely also has antifungal effects on *C. albicans* from other body zones, such as the vagina or nails, in a range close to that which we report in this work.

Interestingly, other Mexican propolis that are not of the poplar type have presented fungicidal effects, such as was the case for those obtained in some other states of Mexico (Veracruz, State of Mexico, and Puebla), presenting antifungal effects against 37 samples of *C. albicans*. One such strain was ATCC 10231, which was used as a reference strain, while the other 36 strains were obtained from human isolates (obtained from: 11 oral exudates, 11 nail flakes, 7 skin flakes, 4 blood, 1 bronchial fluid aspirate, and 1 urine). In this work, the propolis sample obtained from the State of Mexico was more active against the fungus strains, as at a concentration of 800 µg/mL, it inhibited 94.40% of the *C. albicans* human isolate strains, while a concentration of 600 µg/mL was sufficient to inhibit the reference strain. Therefore, the reference strain was more susceptible to all Mexican propolis samples [89,100]. Compared to the other propolis from Mexico, our propolis sample was more active, as the mean range of the FC_50_ was 65 µg/mL for all *C. albicans* strains tested. This variability of anti-*Candida* activity between propolis from the same country is a point to consider, as the desert climatological and environmental characteristics of the considered location differ from the template and tropical climate of the three states mentioned above. This is due to the flora that can be found in these zones differing greatly [101,102,103,104,105], and therefore, the chemical composition and the biological properties of the propolis collected in the different zones of Mexico may have influence on their capacity to inhibit the growth of *C. albicans*.

Among the virulence factors displayed by *C. albicans* are the secretion of hydrolytic enzymes, biofilm formation, and germ tube formation, which are involved in diverse processes linked to the adhesion and formation of biofilms related to the pathological effects of infection by this micro-organism [106]. For this reason, in this work, we evaluate the effect of propolis on the formation of the germ tube. Inhibition of the germ tube is a factor that is directly related to the adhesion process of *C. albicans* with surfaces in organisms; moreover, it is also related to other factors, such as pH, temperature, extracellular enzymatic activities, and glycoproteins. As such, adhesion is considered the first step in the colonization of this micro-organism, leading to the formation of a biofilm [98]. *C. albicans* biofilms are formed of protein, lipids, extracellular DNA, and carbohydrates (which contribute to 25% of the matrix, where the mannan–glucan complex is the most abundant) [107]. Therefore, inhibition of the germ tube is a key mechanism that can determine the antifungal effects displayed by some natural products, such as propolis, on *Candida* species.

In this sense, we investigated the inhibitory effect of Mexican propolis on the germ tube formation of *C. albicans* isolates. Complete inhibition of the germ tube was observed in a range from 1250 µg/mL to 2500 µg/mL of propolis; additionally, the IC_50_ range in which Mexican propolis inhibited the formation of the germ tube ranged from 19 µg/mL to 1290 µg/mL. Similarly, Gomaa and Gaweesh [98] have evaluated the antifungal effect of Egyptian propolis, and showed that, at a concentration of 75 ng/mL (value of MFC), 35% of germ tube formation was reduced, while the MIC value was 100 ng/mL, in a *C. albicans* strain isolated from the oral cavity. Furthermore, Haghdoost et al. [88] have reported that an Iranian propolis had the capacity to inhibit germ tube formation in a concentration-dependent manner, wherein using ¼ MIC reduced it by 22%, while using ½ MIC reduced it by 36.70%. The ability of different propolis samples to inhibit germ tube growth indicates great promise in the search for new antifungals, as this cellular and biochemical process is the initial step for *C. albicans* to begin its behavior as a pathogen, thus developing all its pathogenicity mechanisms.

In addition, in order to find a better explanation of the reported antifungal activities, an analysis of the chemical composition of Mexican propolis was performed, as some authors have suggested that such information is necessary to better understand the biological activities of propolis from different regions [108]. Within the components found in Mexican propolis, we observed a high frequency of compounds of a phenolic origin, such as flavonoids, which have various beneficial properties for human health. The most notable difference from the Iranian propolis [88] was in terms of reports of the chemical composition of the extracts, where several compounds were observed that were absent in the Mexican propolis. The differences in the activities between the two propolis samples were likely due to this parameter, which is determined by the biological properties of propolis from different geographical regions. It is worth mentioning that the chemical analysis reported by Haghdoost et al. [88] did not include abundance values, and so, it is not easy to attribute the antifungal activity to one or more compounds. Likewise, a comparison with the results presented by Gomaa and Gaweesh [98] is difficult, as although there were some similarities in the extract-type used in their work (ethanolic extract) with respect to that which we used (methanolic extract), there was a lack of information about the chemical composition of the Egyptian propolis. Although it is probable that the phenolic compounds in this propolis sample are closely related to their activity on the germ tube growth of *C. albicans*, without knowing the specific compounds presented in the samples, the relation between Mexican propolis and Egyptian propolis can only be inexactly determined.

For this reason, it is important to study the components of propolis, as the inhibitory effect of Mexican propolis on *Candida* sp. can be explained through its chemical composition. Namely, some flavonoids present in our sample have been shown to have antifungal effects, such as pinocembrin [109,110,111,112,113], which has been shown to have an inhibitory effect on *C. albicans* biofilm formation as well as on the filamentous form of this micro-organism. The inhibitory mechanism of pinocembrin is related to a decrease in the cell surface hydrophobicity of *C. albicans* at a concentration of 100 μM, coupled to the fact that, at this same concentration, pinocembrin decreased ALS3 and ACT1 mRNA levels [112]. It is important to keep this last point in mind, as the hydrophobicity alterations could be related to the expression of ALS3 on the cell surface of this fungus; moreover, ALS3 expression is also related to the biofilm formation of *C. albicans* [114,115,116]. On the other hand, ACT1 is a housekeeping gene of *C. albicans* which plays an important role in the hyphal-inducing signals in this micro-organism; therefore, the alteration in the level of ACT1 may also interfere with the hyphal transition of *C. albicans* [117,118]. In addition, the chemical configuration of pinocembrin—especially the hydroxy group at the five position, the ketone group at the four position, and the six-member condensed with benzene—also plays a crucial role in its inhibitory effect on biofilm formation [119].

Like pinocembrin, other flavanones have been shown to possess notable anti-*Candida* effects. Such is the case of naringin and naringenin, whose effects on cellular and biochemical processes related to the death of this fungus have been previously reported [120,121,122,123]. In addition, these two flavanones have also been reported in other Mexican propolis samples [85,87]. In a work carried out by Kim and Lee (2021) [120], it has been reported that the interaction of *C. albicans* with naringin generates cell death by apoptosis in this pathogen, and the authors also confirmed an increase in the production of reactive oxygen species, mitochondrial Ca^2+^ overload and mitochondrial superoxide radical generation, and mitochondrial membrane potential alteration, which generated the detection of the release of cytochrome c from the mitochondria to the cytosol to promote the initiation of apoptosis; moreover, externalization of phosphatidylserine and DNA fragmentation was observed, which are cellular process characteristics of apoptosis by the mitochondrial (intrinsic) pathway. As such, it is very likely that naringin has an important role in the anti-*Candida* effect of Mexican propolis, and further studies should be conducted to determine the type of cell death that it promotes in *C. albicans*.

Similarly, Naringenin is another compound that could be related to the anti-*Candida* effects of Mexican propolis, as there have been reports of the presence of this flavanone in extracts of some plants that have antimicrobial effects against different pathogens, including *C. albicans* [124,125,126]. In addition, naringenin and naringin have been reported to have an anti-*Candida* effect, and are capable of recognizing and binding to enzymes such as topoisomerase II in *Candida* sp. [121]. It should be noted that topoisomerase II plays a crucial role in the dynamics of gene expression, as this enzyme is necessary for the correct unwinding and compaction of DNA and, therefore, for its replication [121]. Considering the above, it is likely that Mexican propolis may have an effect on these enzymes, affecting gene expression and protein synthesis. Furthermore, based on the chemical composition profile, we could suggest that Mexican propolis has an effect on different sites and cellular levels, such as in the nucleus and on mitochondrial function and integrity. The finding regarding the activity on topoisomerase II is of great relevance, as although there have been several reports on flavonoids with this property [122,123], there are few reports on propolis acting at this level, which opens a new gap for the study of this bee product in the context of the inhibition of these enzymes, in order to apply it against various pathogens of medical relevance, and even in diseases such as cancer.

Another compound found in our propolis was baicalein. It has been reported that this flavone is capable of inhibiting biofilm formation in *C. albicans* samples from ATCC in a concentration ranging from 0.0063 to 100 µg/mL [127]. Similarly, baicalein inhibited more than 70% of the biofilm in clinical isolates of *C. albicans* at concentrations between 4 and 32 μg/mL. This flavone was also able to decrease the hydrophobicity of the cell surface of this fungi by decreasing the expression levels of CSH1 mRNA at a concentration of 8 µg/mL [128]. Therefore, it is likely that the anti-biofilm effect of baicalein is also directly related to the inhibition of the germ tube, as these are closely related processes.

Rhamnetin, also identified in our propolis, is a flavonol that has been shown to have an anti-*Candida* effect; however, there are still very few studies on its antimicrobial effects, and in fact, there has only been one study that mentioned that it presents an anti-*Candida* effect [129]. It should be noted that other compounds from the flavonol group, such as kaempferol and quercetin, have demonstrated anti-*Candida* activity [130,131,132,133]. The quercetin MIC values reported for diverse *Candida* sp. are in the range of 0.50–16 μg/mL for *C. parapsilosis* and >128 μg/mL for *C. albicans* and *C. tropicalis*; meanwhile, for kaempferol, the range was from 32–128 μg/mL for *C. parapsilosis*, and from 256–512 μg/mL for *C. albicans* and *C. tropicalis* [130,131,132]. Likewise, both flavonols reduced the metabolic activity and biomass formed by *C. parapsilosis* complexes; nevertheless, kaempferol was more effective than quercetin [130]. Specifically, kaempferol has been reported to have an MIC against *Candida* sp. ranging from 256–512 μg/mL; in particular, this flavonoid had a synergic antifungal activity in *C. tropicalis* through the histone deacetylase (HDAC) inhibitor [133]. Kaempferol has also displayed inhibitory effects on fluconazole-resistant *C. albicans* strains (with an MIC ranging from 128–256 μg/mL), where the mechanism related to the inhibitory effect of this flavonoid was related to a reduction in the expressions of the genes CDR1, CDR2, and MDR1 [131], which are genes related to fluconazole resistance in *C. albicans* through controlling the up-regulation of the multi-drug efflux pump [134,135,136].

On the other hand, flavonoids from other families, such as flavan-3-ols, have also presented anti-*Candida* properties, including catechin [137,138,139]. This compound inhibits *C. albicans* proliferation through the suppression of ATP production and the inhibition of the hyphal transformation; in this latter point, catechin at a 800 µg/mL concentration (MIC value reported for *C. albicans*) was found to reduce the mRNA expressions of the hypha-specific genes HWP1, ALS3, SAP4, SAP5, SAP6, CPH1, and EFG1; at the same time, the gene RAS1 and the yeast-specific gene YWP1 were induced. Therefore, at this concentration, catechin displayed a fungistatic effect on *C. albicans*. Moreover, the suppression of hyphal-specific genes suggests that catechin suppresses the pathogenicity effect of *C. albicans* through the inhibition of hyphal formation. Additionally, the disruption of Cek1 phosphorylation displayed by catechin suggests that this flavonoid interferes with the MAP kinase cascade and cAMP pathway [137], both of which are closely related to *C. albicans* hyphal formation [138,139]. Due to the wide floral and plant diversity of Mexico, it is very likely that propolis from other geographical regions contain flavonoids such as catechin, and even many other flavonoids or secondary metabolites with antifungal and antimicrobial properties; therefore, formal research should continue to be carried out using various Mexican propolis samples.

Although many works highlight the presence and activity of phenolic compounds such as flavonoids, they are not the only molecules that may have beneficial medicinal properties for health. Recently, cycloartane-type triterpenic acids isolated from Cameroonian propolis were shown to have potent antimicrobial activity and are specifically capable of inhibiting biofilm formation in *C. albicans* (22.30–40.10%) and *C. tropicalis* (13.50–44.50%). Something remarkable is that in this work, the pure triterpenic cycloartane type showed greater activity than the total extract of propolis, so we can consider that there are antagonist relationships in the entire complex mixture of said extract [140].

Another interesting aspect is that stingless bee (Meliponini, Apidae) propolis from the Kilimanjaro area, Ngarony locality, Tanzania, also showed outstanding antimicrobial properties and, specifically, can inhibit *C. albicans* biofilm formation. However, little is known about the benefits of propolis for stingless bees as its study is relatively recent. Like the bees of the genus *Apis*, the bee products of the stingless bees of Tanzania have a different chemical composition than that of propolis from Europe (Belgium) [141]. This information shows that both the propolis of the bees of the genus *Apis* and those of the Meliponini tribe are products that deserve to be studied and considered as an option in the search for new therapeutic options.

Nevertheless, these results only can explain a part of the possible mechanisms of action involved in the antifungal effect of Mexican propolis. It is important to consider other inhibitory mechanisms related to *Candida* sp., such as membrane damage, the inhibition of acid nucleic synthesis, and the inhibition of energy metabolism, as well as the morphological change from yeast to hyphae (known as yeast–hyphae dimorphism) [142,143]. At this point, the variability and complexity in the chemical constituents present in Mexican propolis makes it a product with a very interesting study potential, allowing those different phenolic compounds to exert their antifungal effects through synergism among them, contributing to the inhibitory effects on micro-organism growth and to the inhibition of virulence factors related to infection with *Candida* sp. [105].

## 4. Materials and Methods

### 4.1. Obtaining and Preparing Propolis

The sample of Mexican propolis was collected at the apiary of Mr. José Luis Gonzalez, located in the municipality of Chihuahua, Chihuahua, Mexico, in December 2018. For the methanolic extract, 335 g of propolis were macerated with 1 L of methanol (99.8%) (Sigma-Aldrich, St. Louis, MO, USA). This process was carried out several times (once a week for 7 weeks) until a very light tone was observed in the maceration. The yield of the extract was 216.64 g, equivalent to 64.67% of the propolis placed to macerate. The propolis extract was stored at 4 °C in the dark.

### 4.2. Collection of Clinical Samples of Candida

Ten *Candida* samples were donated by the Clínica Odontológica Periférica of the Facultad de Estudios Superiores Iztacala (FES-Iztacala, Tlalnepantla, Estado de México, México), whose headquarters is in the FES-Aragón (Nezahualcóyotl, Estado de México, México) of the Universidad Nacional Autónoma de México (UNAM). The samples were obtained from clinical cases with suspected *Candida* infections by smearing the injured sites of the tongue in the oral cavity with a sterile cotton swab, with the informed consent of the patients. Afterwards, each of the oral samples obtained were placed in Petri dishes containing PDA (Potato Dextrose Agar; Becton Dickinson, Estado de México, México.) and incubated for 48 h at 37 °C for growth.

### 4.3. Candida Species Identification

*Candida* samples from Petri dishes containing PDA were subsequently placed in Petri dishes containing CHROMagar^TM^ *Candida* culture medium (BBL^TM^ CHROMagar^TM^ *Candida* Medium; Becton Dickinson, Guadalajara, Jalisco, México.), which was used as a chromogenic medium for the presumptive differential characterization of *C. albicans*, *C. tropicalis*, *C. krusei*, and *C. glabrata* by evaluating light-to-medium green, blue-greenish to metallic-blue, light rose, and cream colonies, respectively, which characterize these yeasts, as indicated in the manufacturer’s guidelines. The samples were incubated at 37 °C for 48 h for growth and subsequent identification.

### 4.4. Anti-Candida Activity Qualitative and Quantitative Assays

The qualitative anti-*Candida* activity of Mexican propolis was determined by the disk diffusion method [144]. Filter paper discs (diameter 5 mm) were impregnated with 10 mg of propolis, 25 µg of nystatin (positive control; Sigma-Aldrich), or 10 µL of methanol (negative control). The different species of *Candida* were cultured for growth in 10 mL of Sabouraud broth (Becton Dickinson, USA) at 37 °C for 48 h before interaction with the propolis, in order to subsequently inoculate Petri dishes containing PDA by submerging a sterile cotton swab in a standard suspension of 1 × 10^6^ CFU/mL of the cultures of each *Candida* sp. and seed it uniformly over the entire surface of the agar. Finally, the different discs (previously impregnated) were placed in triplicate in the inoculated agar, and the dishes were left in an incubator for 48 h at 37 °C. After incubation, the diameters of the inhibition zone presented by the tested substances were measured in millimeters (mm), which are reported as inhibition halos.

The quantitative anti-*Candida* activity of Mexican propolis was determined by the broth dilution microtechnique [145]. The different species of *Candida* were cultured for their growth in 10 mL of Sabouraud broth at 37 °C for 48 h before interaction with the propolis. Subsequently, in a 96-well plate, 150 µL per well of the different concentrations of the propolis (i.e., 15 serial concentrations from 20,000 to 0.0010 µg/mL) or nystatin (i.e., 7 serial concentrations from 10 to 0.1560 µg/mL) to be tested were added (in triplicate). Then, 50 µL of the standard suspension (1 × 10^6^ CFU/mL) of each cultured *Candida* were added to the wells, following which, the plate was incubated for 48 h at 37 °C. After this time, a sample was taken from each well and seeded in septated Petri dishes containing PDA agar, which were incubated for 48 h at 37 °C. Subsequently, we counted the CFUs with respect to each of the propolis concentrations tested. Finally, the Minimum Fungicide Concentration (MFC), the 75% Fungicide Concentration (FC_75_), and the Medium Fungicidal Concentration (FC_50_) of the propolis were determined. Concentrations are reported in µg/mL.

### 4.5. Candida Germ Tube Growth Inhibition Assay

The germ tube growth inhibition assay [146] was performed only with samples of clinical isolates identified as *C. albicans*, which were cultured for growth in 10 mL of Sabouraud broth at 37 °C for 48 h before interaction with the Mexican propolis. Based on the MFC, FC_50_, and FC_25_ determined in the quantitative assay of the anti-*Candida* activity of propolis detailed above, different serial concentrations (from 2500 to 312 µg/mL) of MFC were tested. To test the FC_50_ and FC_25_, twice as much was used, with respect to the concentration determined in the quantitative assay. Fetal bovine serum (Gibco, USA) was added to the microcentrifuge tubes, then an inoculum of the standard suspension (1 × 10^6^ CFU/mL) of the culture grown for each *Candida* was aggregated, and the different concentrations of propolis to be tested were immediately added. Finally, the tubes were incubated for 4 h at 37 °C. Tests were performed in triplicate. After the interaction, an aliquot was taken from each tube to count the yeasts with germinative tube growth (control germ tube) in the Neubauer chamber. DMSO (less than 0.01%) was used as a negative control. The 50% inhibitory concentration (IC_50_) of propolis was determined. Concentrations are reported in µg/mL.

### 4.6. General Experimental Procedures and Equipment Utilized for the Determination of the Chemical Composition of Mexican Propolis

Nuclear Magnetic Resonance (NMR) spectra were collected using a Varian NMR System (Yarton, Oxford, England) at 500 MHz and Bruker spectrometers (Bruker Daltonics, Ettlingen, Germany) at 600 and 700 MHz using chloroform-d, methanol-d4, acetone-d6, and dimethyl sulfoxide-d6 as solvents and with tetramethylsilane as an internal standard. Thin-layer chromatograms were performed on pre-coated thin-layer chromatography TLC sheets of silica gel Merck 60-F254. Spots on TLC were visualized under a UV lamp or developed by being sprayed with cerium molybdate and cerium sulfate. Fractionation of the extract was performed by open column chromatography on silica gel (Merck 230–400 mesh). High Resolution Mass Spectra (HRMS) were determined using a Bruker microTOF-QII spectrometer (Bruker Daltonics, Billerica, MA, USA), with desorption electrospray ionization mass in negative mode and a constant volumetric flow rate (8 µL/min). The capillary voltage was set to 3500 V, and nitrogen was used as the drying and nebulizing gas, with a flow rate of 0.4 Bar (4.0 L/min) and a gas temperature of 180 °C. Mass spectra were collected in the *m*/*z* range of 50 to 3000. Data were processed using the Bruker Compass Data Analysis 4.0 software (Bruker Daltonics, Billerica, MA, USA), while identification of compounds based on MS measurements was performed using the Compound Crawler 3.0 software (Bruker Daltonics, Billerica, MA, USA).

#### Fractionation and Identification of Chemical Constituents from Propolis Methanolic Extract

The propolis methanolic extract was monitored by ^1^H NMR, and subsequently, 1 g was subjected to fractionation by column chromatography in a hexane:dichloromethane:ethyl acetate (Hex:CH_2_Cl_2_:AcOEt) system (stationary phase: silica gel flash; fraction volume: 50 mL; elution system: 6:3:1). We obtained 189 fractions, which were analyzed by TLC. Fractions F40, F75, F80, F101, F130, and F160 were selected and analyzed by ^1^H and ^13^C NMR in order to determine the presence of phenolic compounds and flavonoids. To complement the chemical analysis, the fractions were analyzed through mass spectrometry using the electrospray ionization method in negative mode (ESI-MS). Identification of compounds was supported through direct comparison of their spectral data to those reported in the literature [86].

### 4.7. Statistical Analysis

All data are expressed as the mean ± standard deviation (SD). To determine the MFC, FC_75_, FC_50_, and IC_50_ of the propolis, logarithmic regression analysis was performed using the GraphPad Prism software version 9.3.1 (GraphPad Software Inc., San Diego, CA, USA). Concentrations are reported in µg/mL. For qualitative and quantitative activity data, a two-way ANOVA statistical analysis was performed, followed by Tukey’s test. *p* < 0.05 was considered significant.

## 5. Conclusions

The present work is the first to demonstrate the antifungal effect of propolis from northern Mexico (Chihuahua state) against clinical isolates of *Candida* sp. and, in the same way, the inhibition of germ tube formation after the interaction of *C. albicans* with the propolis extract. The propolis chemical composition was characterized by a high content of compounds of s phenolic origin, where the presence of pinocembrin, baicalein, and rhamnetin indicated that this propolis from northern Mexico presents anti-*Candida* effects. This work demonstrates the importance of studying the products used in alternative and traditional medicine in order to support their use in a scientific and safe manner. Future works should be focused on research on the content of Chihuahua propolis extract’s secondary metabolites and their implications for the action mechanisms involved in the antifungal effect of this bee-derived natural product, with the aim to develop a better human and veterinary treatment of mycosis.

## Figures and Tables

**Figure 1 molecules-27-05651-f001:**
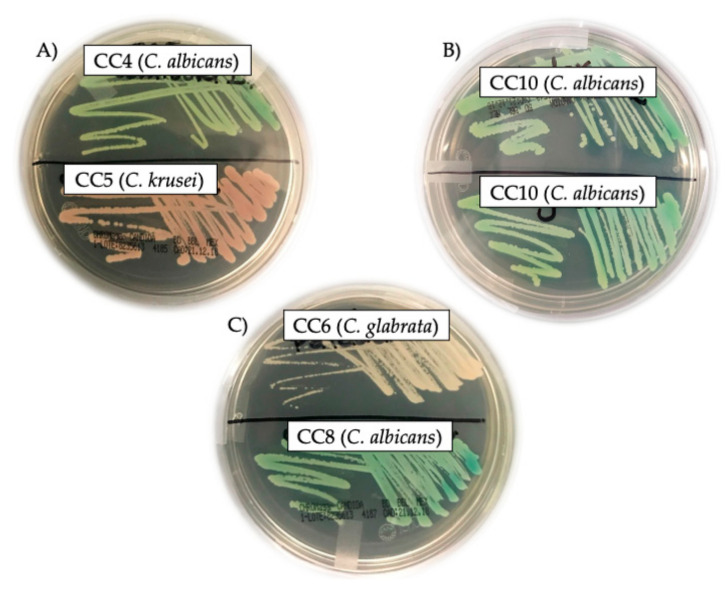
Representative photographs of the identification of the different species of *Candida*: (**A**) Upper half of Petri dish is CC4 (*C. albicans*), lower half of Petri dish is CC5 (*C. krusei*); (**B**) Upper and lower halves of Petri dish are CC10 (*C. albicans*); (**C**) Upper half is CC6 (*C. glabrata*), lower half is CC8 (*C. albicans*).

**Figure 2 molecules-27-05651-f002:**
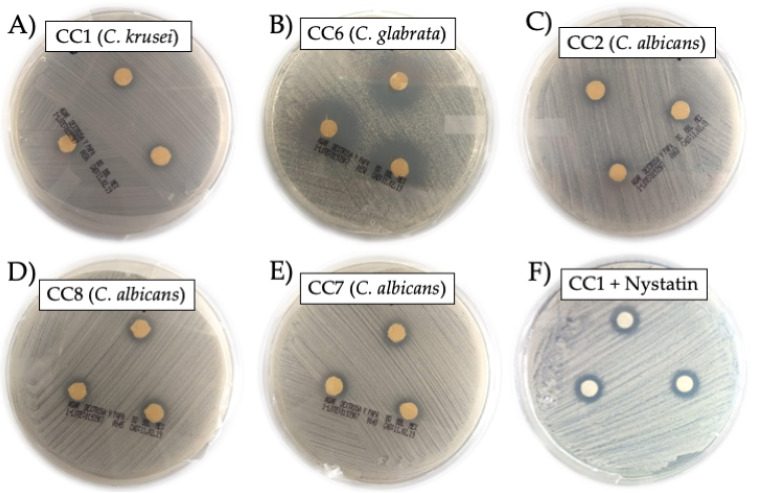
(**A**–**E**) Representative photographs of the inhibition halos of the antifungal activity of Mexican propolis against different species of *Candida*. (**F**) Representative photograph of the inhibition halos of Nystatin against CC1 (*C. krusei*).

**Figure 3 molecules-27-05651-f003:**
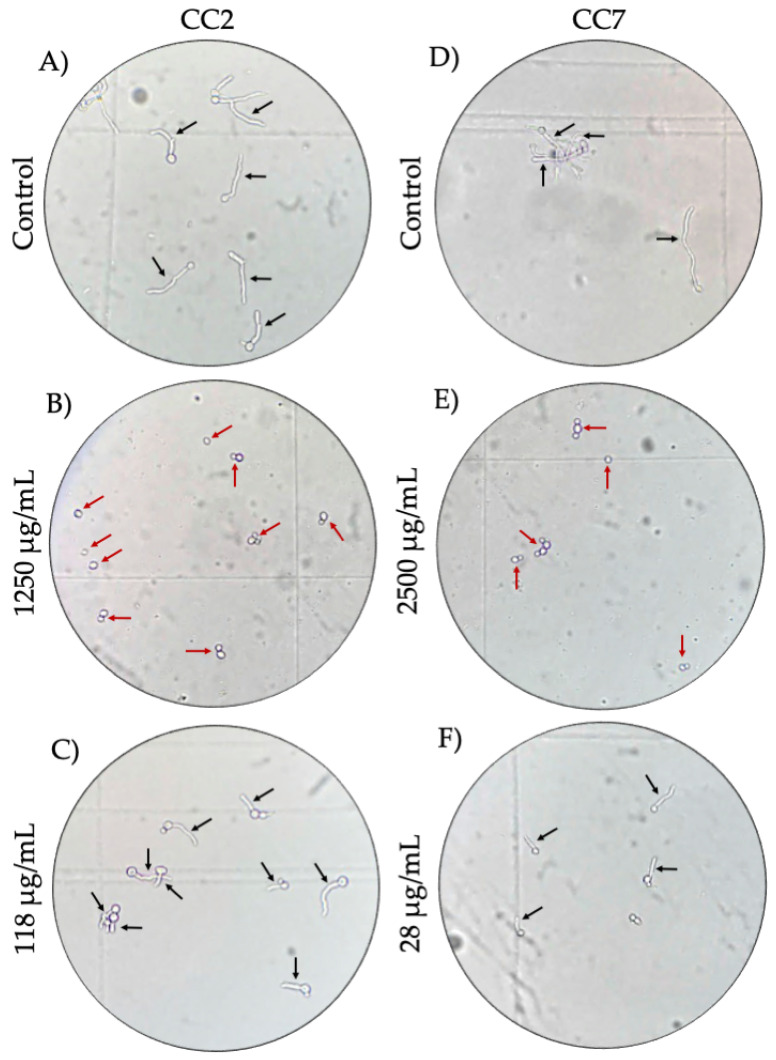
Representative microphotographs (40×) of the inhibitory effect of propolis on the germ tube growth of *C. albicans*. Black arrows point to the germ tubes of *C. albicans*. Red arrows point to *C. albicans* without germ tube growth. (**A**,**D**) are the germ tube growth of *C. albicans* from CC2 and CC7, respectively, after four hours of culture. (**B**) Complete inhibition of the germ tube in CC2 with a concentration of 1250 µg/mL of Mexican propolis. (**C**) Germ tube growth inhibition of 51.54% in CC2 at 118 µg/mL of propolis. (**E**) Complete inhibition of the germ tube at 2500 µg/mL in CC7. (**F**) Germ tube growth inhibition of 60.08% in CC7 at 28 µg/mL of propolis.

**Figure 4 molecules-27-05651-f004:**
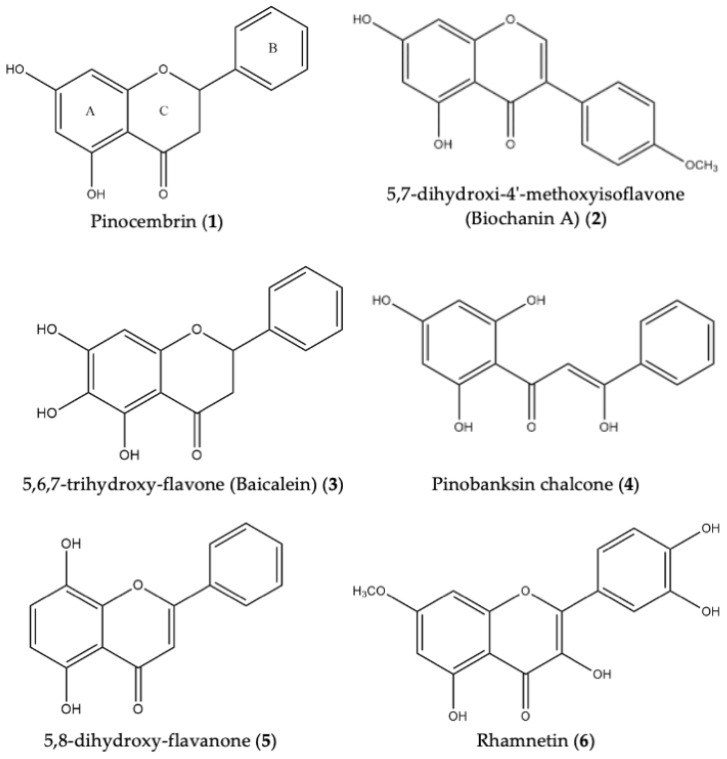
Structural formulas of flavonoids present in Mexican propolis.

**Table 1 molecules-27-05651-t001:** Qualitative evaluation of the anti-*Candida* activity of Mexican propolis.

*Candida* Strain	Sample	Propolis (mm)	Nystatin (mm)
*C. krusei*	CC1	8.83 ± 0.11	9.7 ± 0.58
	CC5	7.60 ± 0.10	18.0 ± 0.98
	CC9	8.82 ± 0.11 ^f^	19.5 ± 0.87 ^f^
*C. albicans*	CC2	9.80 ± 0.26	19.0 ± 1.00
	CC3	9.50 ± 0.34	20.1 ± 1.00
	CC4	8.83 ± 0.28	18.7 ± 1.15
	CC7	9.80 ± 0.10 ^f^	19.5 ± 0.45 ^f^
	CC8	9.56 ± 0.55 ^f^	18.4 ± 1.21 ^f^
	CC10	10 ± 1 ^f^	21.3 ± 1.01 ^f^
*C. glabrata*	CC6	21.43 ± 1.30 ^a, b, c, d, e^	36.7 ± 1.88 ^a, b, c, d, e^

Inhibition halos are reported in millimeters (average data from three replicates). The Mexican propolis was tested at a concentration of 10 mg per disc. All values are expressed as the mean ± SD. (^a^) compared with CC1 group; (^b^) compared with CC2 group; (^c^) compared with CC3 group (^d^) compared with CC4 group; (^e^) compared with CC5 group; (^f^) compared with CC6 group. In all cases, the value of *p* was < 0.05.

**Table 2 molecules-27-05651-t002:** Quantitative evaluation of the anti-*Candida* activity of Mexican propolis.

*Candida* Strain	Sample	Propolis			Nystatin		
		MFC	FC_75_	FC_50_	MFC	FC_75_	FC_50_
*C. krusei*	CC1	625 *	195 ± 0.0135	107 ± 0.0075	5	3.79 ± 0.0012	2.56 ± 0.0015
	CC5	1250 *	331 ± 0.0150	244 ± 0.0080	10	6.70 ± 0.0023	4.11 ± 0.0015
	CC9	625 *	234 ± 0.0185	109 ± 0.0110	5	3.33 ± 0.0017	2.05 ± 0.0012
*C. albicans*	CC2	312 *	98 ± 0.0215	59 ± 0.0142	2	1.81 ± 0.0010	1.19 ± 0.0012
	CC3	312 *	75 ± 0.0165	48 ± 0.0090	2	1.63 ± 0.0006	1 ± 0.0015
	CC4	1250 *	492 ± 0.0220	237 ± 0.0195	10	7.37 ± 0.0020	4.85 ± 0.017
	CC7	1250 *	19 ± 0.0067	14 ± 0.0031	10	5.51 ± 0.0021	3.14 ± 0.0020
	CC8	625 *	28 ± 0.0045	18 ± 0.0020	5	2.69 ± 0.0023	1.54 ± 0.0021
	CC10	1250 *	21 ± 0.0030	15 ± 0.0015	10	6.55 ± 0.0025	3.96 ± 0.0017
*C. glabrata*	CC6	312 *	59 ± 0.0131	30 ± 0.0055	2	1.53 ± 0.0020	0.92 ± 0.0010

MFC, FC_75_, and FC_50_ are reported in µg/mL. All values are expressed as the mean ± SD. (*) *p* < 0.05 compared to the nystatin group.

**Table 3 molecules-27-05651-t003:** Effect of Mexican propolis on the inhibition of germ tube growth of *C. albicans*.

*C. albicans.*	IC_50_
CC2	108 ± 0.0021
CC3	138 ± 0.0012
CC4	1291 ± 0.0141
CC7	33 ± 0.0006
CC8	30 ± 0.0010
CC10	19 ± 0.0015

IC_50_ are reported in µg/mL.

## Data Availability

The data presented in this study are available on request from the corresponding author.

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
