# Peer review of "Antifungal Activity of Mexican Propolis on Clinical Isolates of Candida Species"

_molecules, 2022, doi:10.3390/molecules27175651_

Round 1
Reviewer 1 Report
Review [Molecules] Manuscript ID: molecules-1874466
Some specific comments and observations about the review are as follows:
1. Figure 3 should be reviewed because its quality is inadequate. The size of the images is very small, and it is therefore difficult to see the modifications of the images in detail. Perhaps, it would be appropriate to make the image bigger if that is possible.
2. Tables 2 and 3.- the data are given in mg/ml and in the antiinfectives the IC50 and MICs are usually shown in ug/ml. Therefore, it would be advisable to unify the comparations with the previous data in order to show the potency of the propolis extract.
3. The introduction and discussion should be completed with data of natural products from microorganism sources because they are other examples for getting compounds from natural products. Fundacion MEDINA has different examples of this.
4. The information of the clinical isolates should be completed with their sensitivity to different antibiotics in order to improve the robustness of the results considering that, perhaps, they have some resistance and so this would give more value to the effect of the propolis extracts.
5. The compounds isolated and identified in the extracts should be tested against the Candida sp. to complete the study and be sure that they are in fact responsible for the effect observed with the extracts.
6. It would be of great value to include an ATCC strain in the results if this is possible to compare them with the clinical isolates.
7. In the discussion there is a 37 C without the “º”. The latter must be added.
8. In point 4.5, tube growth inhibition assay, in the section on material & methods, the controls used in the assay should be indicated.
9. Future work needed to be done with the results of the article should be added to the conclusions.
10. All references should be reviewed in general to bring them into line with the standard format.
11. There should also be a general review of the English.
Author Response
Reviewer 1
Some specific comments and observations about the review are as follows:
- Figure 3 should be reviewed because its quality is inadequate. The size of the images is very small, and it is therefore difficult to see the modifications of the images in detail. Perhaps, it would be appropriate to make the image bigger if that is possible.
Reply:
Figure 3 corrected in the text.
- Tables 2 and 3.- the data are given in mg/ml and in the antiinfectives the IC50 and MICs are usually shown in ug/ml. Therefore, it would be advisable to unify the comparations with the previous data in order to show the potency of the propolis extract.
Reply:
Corrected in the text and Table 1 and 2.
- The introduction and discussion should be completed with data of natural products from microorganism sources because they are other examples for getting compounds from natural products. Fundacion MEDINA has different examples of this.
Reply:
The suggestion complements the discussion in a good way, and we even started this section with this topic. In the line 221 the following paragraph was added:
“Natural products are an invaluable source of compounds with various medicinal properties. These bioactive molecules come from various sources, including terrestrial plants, microorganisms, marine organisms, and terrestrial vertebrates and invertebrates. Microorganisms have provided great contributions such as penicillin and other antimicrobials. Similarly, compounds of plant origin provide a host of new agents with therapeutic potential. Propolis is a bee product made by bees from compounds of plant origin and has proven to be a great candidate for treating different health conditions.”
- The information of the clinical isolates should be completed with their sensitivity to different antibiotics in order to improve the robustness of the results considering that, perhaps, they have some resistance and so this would give more value to the effect of the propolis extracts.
Reply:
The suggestion is interesting, considering the resistance of different microbial strains to different drugs. However, we consider that testing clinical isolates with other drugs could be part of another later work where the focus is to determine the effects of propolis compared to different treatments currently used against different mycoses. Furthermore, the focus of the work is to set a precedent that favors the use of Mexican propolis and the arid zones of the American continent, so it was not considered in the objectives and scope of this work include that essay. For these reasons we consider that for this work we will not include that essay. Regarding this suggestion, in line 563 we add the following text as a perspective of our working group:
“In future studies we suggest comparing the effect of propolis with different drugs used to treat fungal infections, it could even be interesting to determine the effect of propolis in combination with some of these drugs.”
- The compounds isolated and identified in the extracts should be tested against the Candida sp. to complete the study and be sure that they are in fact responsible for the effect observed with the extracts.
Reply:
The proposal is very complex. So far, our working group has been able to identify through Nuclear Magnetic Resonance (NMR), column chromatography, and High-Resolution Mass Spectra (HRMS) analysis, some of the compounds present in Mexican propolis. Unfortunately, in our work did not be isolated any compound. We continue the processing of propolis for the purification and isolation of the compounds, but it is an arduous process that requires more time. For this reason, it is impossible for us to test the pure compounds.
In our perspectives as a working group, in the future researches we intend focus to use the isolated compounds of Mexican propolis on C. albicans, but we must propose an experimental strategy that allows us to optimize the use of the limited quantities that are normally obtained of pure compounds.
- It would be of great value to include an ATCC strain in the results if this is possible to compare them with the clinical isolates.
Reply:
In the experimental design we did not include an ATCC strain since by using strains obtained from clinical cases, we are testing propolis on pathogens that are capable of infecting and invading an individual. In addition, since many works carried out with propolis and natural products use ATCC strain for their tests, we consider that using clinical isolates gives a beneficial and original point to our work. For this reason, we consider that we can preside over that strain.
- In the discussion there is a 37 C without the “º”. The latter must be added.
Reply:
The number 37 refers to the number of samples of C. albicans. To avoid confusion, it was corrected in line 286:
“...against 37 samples of C. albicans...”
- In point 4.5, tube growth inhibition assay, in the section on material & methods, the controls used in the assay should be indicated.
Reply:
In line 520 of section 4.5 of materials and methods the following text was added:
“…germinative tube growth (control germ tube) in the Neubauer chamber. DMSO (less than 0.01%) was used as a negative control.”
- Future work needed to be done with the results of the article should be added to the conclusions.
Reply:
In the conclusions section the following text was added on line 565:
“Future works should be focused on the research of Chihuahua propolis extract secondary metabolites contented and their implication in the action mechanisms involved in the antifungal effect of this bee-derived natural product, with the aim to develop a better human and veterinary treatment of mycosis.”
- All references should be reviewed in general to bring them into line with the standard format.
Reply:
The references were reviewed, corrected and added with the parameters requested by the journal.
- There should also be a general review of the English.
Reply:
The language was corrected by MDPI English editing service, with certificate number 47894.

Reviewer 2 Report
Dear Editor,
Molecules MDPI.
Thank you very much for giving me the opportunity to review the manuscript ‘Antifungal Activity of Mexican Propolis on Clinical Isolates of Candida Species’
The article is very significant as it reports the application of propolis to combat microbial infections caused by candida species. Propolis is used principally as a chemical weapon to keep the hive free from microbes and therefore testing different propolis samples on pathogenic microbes is always interesting information.
The authors can address the following issues so as to improve on the nature of their paper.
1. Complete the citation in the template…..Citation: Lastname, F.; Lastname, F.; Lastname, F. Title. Molecules 2022, 27, x. https://doi.org/10.3390/xxxxx
2. In the introduction, the authors mentioned virulence factors. I recommended that the authors should use these two papers concerning propolis to improve this aspect and also their discussion
https://doi.org/10.3390/molecules27154872: Antibiofilm and Anti-Quorum Sensing Potential of Cycloartane-Type Triterpene Acids from Cameroonian Grassland Propolis: Phenolic Profile and Antioxidant Activity of Crude Extract
https://doi.org/10.3390/foods10050997: A Preliminary Study of Chemical Profiles of Honey, Cerumen, and Propolis of the African Stingless Bee Meliponula ferruginea
3. The English language should be improved.
4. Harmonize values to have the same number of places after decimal point…eg line 130.
5. In figure 2, authors should include control plates.
6. The authors should improve results on FC50… saying highest FC50 is very ambiguous because, the lower the FC50 the higher the activity. I recommend that authors should use different words from highest. Same for IC50.
7. Format Paragraph 4.6.1
Author Response
Reviewer 2
Thank you very much for giving me the opportunity to review the manuscript ‘Antifungal Activity of Mexican Propolis on Clinical Isolates of Candida Species’
The article is very significant as it reports the application of propolis to combat microbial infections caused by candida species. Propolis is used principally as a chemical weapon to keep the hive free from microbes and therefore testing different propolis samples on pathogenic microbes is always interesting information.
The authors can address the following issues so as to improve on the nature of their paper.
- Complete the citation in the template…..Citation: Lastname, F.; Lastname, F.; Lastname, F. Title. Molecules 2022, 27, x. https://doi.org/10.3390/xxxxx
Reply:
This information is completed by the journal until the end of the process.
- In the introduction, the authors mentioned virulence factors. I recommended that the authors should use these two papers concerning propolis to improve this aspect and also their discussion
https://doi.org/10.3390/molecules27154872: Antibiofilm and Anti-Quorum Sensing Potential of Cycloartane-Type Triterpene Acids from Cameroonian Grassland Propolis: Phenolic Profile and Antioxidant Activity of Crude Extract
https://doi.org/10.3390/foods10050997: A Preliminary Study of Chemical Profiles of Honey, Cerumen, and Propolis of the African Stingless Bee Meliponula ferruginea
Reply:
The proposed articles are interesting and bring an important perspective to the discussion. In the line 442, the following paragraph was added:
Although many works highlight the presence and activity of phenolic compounds such as flavonoids, they are not the only molecules that may have beneficial medicinal properties for health. Recently, cycloartane-type triterpenic acids isolated from Cameroonian Propolis were shown to have potent antimicrobial activity and are specifically capable of inhibiting biofilm formation of C. albicans (22.3-40.1%) and C. tropicalis (13.5-44.5%). Something remarkable is that in this work the pure triterpenic cycloartane-type showed greater activity than the total extract of propolis, so we can consider that there are antagonism relationships in the entire complex mixture of said extract.
Another interesting aspect is that stingless bee (Meliponini, Apidae) propolis from the Kilimanjaro area, Ngarony locality, Tanzania also showed outstanding antimicrobial properties and specifically can inhibit C. albicans biofilm formation. However, little is known about the benefits of propolis for stingless bees as its study is relatively recent. Like the bees of the genus Apis, the bee products of the stingless bees of Tanzania have a different chemical composition than that of propolis from Europe (Belgium). This information shows that both the propolis of the bees of the genus Apis and those of the Meliponini tribe are products that deserve to be studied and desired as an option in the search for new therapeutic options.
- The English language should be improved.
Reply:
The language was corrected by MDPI English editing service, with certificate number 47894.
- Harmonize values to have the same number of places after decimal point…eg line 130.
Reply:
Corrected in all text.
- In figure 2, authors should include control plates.
Reply:
Corrected and added in Figure 2.
- The authors should improve results on FC50… saying highest FC50 is very ambiguous because, the lower the FC50 the higher the activity. I recommend that authors should use different words from highest. Same for IC50.
Reply:
In the line 143, the following text was corrected and added:
“…different concentrations of inhibition. The CC2 and CC3 (both C. albicans) and CC6 isolates (corresponding to C. glabrata), presented an MFC of 312 µg/mL. In contrast, CC4, CC7, and CC10 (C. albicans), and CC5 (C. krusei) reported an MFC of 1250 µg/mL. In addition, the most sensitive sample to propolis was C. albicans from CC7 exhibited an FC75 of 19 ± 0.0067 µg/mL and an FC50 of 14 ± 0.0031 µg/mL. In the same sense, CC4 and CC5 were the least susceptible samples to propolis, presenting an FC75 of 492 ± 0.0220 µg/mL and an FC50 of 244 ± 0.0080 µg/mL, respectively. The results…”
- Format Paragraph 4.6.1
Reply:
The instructions for authors of the journal "Molecules" were reviewed, as well as the latest articles published on its website and the format is correct since section 4.6.1 is a subsection of section 4.6.

Reviewer 3 Report
The manuscript "Antifungal Activity of Mexican Propolis on Clinical Isolates of Candida Species" presents interesting research results. The search for new antifungal agents has been the subject of much research, and propolis has the potential to find such an application.
The research is well planned, the discussion of the results is extensive and detailed, the literature is well selected, and most of it is articles from the last 5 years.
Detailed comments:
line 27, 76, 305 - should be microorganisms.
Chapter Introduction - Please provide more detailed information on the antifungal activity of propolis.
Chapter 2.1 - Describe more precisely where Candida was isolated from.
Table 1 - no statistical analysis, no units, the number of decimal places should be even, it would be easier to read the results if the strains in the table were arranged by species.
Needlessly, the same results are presented in Table 1 and Figure 2B.
Table 2 - no statistical analysis.
Chapter 2.4 - the main chemical components identified in propolis are not listed in the table.
Chapter 4.1 - Why did maceration take so long? What concentration of methanol was used?
Chapter 4.3 - Were the strains deposited in the public collection?
Author Response
Reviewer 3
The manuscript "Antifungal Activity of Mexican Propolis on Clinical Isolates of Candida Species" presents interesting research results. The search for new antifungal agents has been the subject of much research, and propolis has the potential to find such an application.
The research is well planned, the discussion of the results is extensive and detailed, the literature is well selected, and most of it is articles from the last 5 years.
Detailed comments:
- line 27, 76, 305 - should be microorganisms.
Reply:
Corrected in the line 28, 77, 250, 306, 311, 365, 354 and 452.
- Chapter Introduction - Please provide more detailed information on the antifungal activity of propolis.
Reply:
In the line 105, the following paragraph was added:
“In this sense, it has been reported that propolis samples from different countries such as Poland, Iran, Cameroon, Brazil, the Czech Republic, Ireland, and Germany present both qualitative and quantitative antifungal activity differently on reference Candida strains such as C. albicans (ATCC 10231, 90028, 66396; CBS 562; NR 29450; SC 5314), C. krusei (ATCC 6258, 90878; CBS 573), Candida parapsilosis (ATCC 22019; CBS 604), and C. glabrata (CBS 07; DSM 11226 ; LMA 90-1085), Candida tropicalis (ATCC 9968; CBS 94) and Candida dubliniensis (CBS 7987), and on clinical isolates of Candida obtained from smears of the mouth, throat, fluid from the peritoneal cavity, bronchopulmonary lavage, stoma, blood , urine, feces, and anus identified as C. albicans, C. glabrata, C. krusei, C. dubliniensis, C. tropicalis, and C. parapsilosis.”
- Chapter 2.1 - Describe more precisely where Candida was isolated from.
Reply:
In the line 118, the following text was added:
“All Candida sp. were isolated from the tongue of the patients.”
And in the line 467, the following text was added:
“…of the tongue…”
- Table 1 - no statistical analysis, no units, the number of decimal places should be even, it would be easier to read the results if the strains in the table were arranged by species.
Reply:
Regarding “no statistical analysis”
In line 557 of section 4.7 of materials and methods the following text was added:
“For qualitative and quantitative activity data, a two-way ANOVA statistical analysis was performed, followed by the Tukey test. P < 0.05 was considered significant.”
And in the line 136, the following text was added:
“All values are expressed as the mean ± SD. (a) compared with CC1 group; (b) compared with CC2 group; (c) compared with CC3 group (d) compared with CC4 group; (e) compared with CC5 group; (f) compared with CC6 group. In all cases the value of P was < 0.05.”
Regarding “no units”
The units of the results in Table 1 are millimeters and are specified at the end of the Table and in line 492 of section 4.4 of materials and methods. But to avoid confusion, the units in the propolis and nystatin column of Table 1 were added.
Regarding “the number of decimal places should be even”
Corrected in all text.
Regarding “it would be easier to read the results if the strains in the table were arranged by species”
Corrected in the Table 1 and 2.
- Needlessly, the same results are presented in Table 1 and Figure 2B.
Reply:
You are right. We removed Figure 2B in the text.
- Table 2 - no statistical analysis.
Reply:
In line 557 of section 4.7 of materials and methods the following text was added:
“For qualitative and quantitative activity data, a two-way ANOVA statistical analysis was performed, followed by the Tukey test. P < 0.05 was considered significant.”
And in the line 152, the following text was added:
“All values are expressed as the mean ± SD. (*) P < 0.05 compared to the nystatin group.”
- Chapter 2.4 - the main chemical components identified in propolis are not listed in the table.
Reply:
We identified 6 compounds in Mexican propolis, so we did not add a table in the text. We considered that it was more attractive for the reader to place the compounds in a image, such as Figure 4.
- Chapter 4.1 - Why did maceration take so long? What concentration of methanol was used?
Reply:
Regarding “Why did maceration take so long?”
To obtain an optimal extraction of propolis, we rely on the maceration technique that has been described in more detail by some research groups that work with propolis extracts from different countries, such as doi: 10.12980/APJTB.4.2014APJTB-2013-0039 (Kustiawan, P. M., Puthong, S., Arung, E. T., & Chanchao, C. (2014). In vitro cytotoxicity of Indonesian stingless bee products against human cancer cell lines. Asian Pacific journal of tropical biomedicine, 4(7), 549-556), DOI:http://dx.doi.org/10.7314/APJCP.2015.16.15.6581 (Kustiawan, P. M., Phuwapraisirisan, P., Puthong, S., Palaga, T., Arung, E. T., & Chanchao, C. (2015). Propolis from the stingless bee Trigona incisa from East Kalimantan, Indonesia, induces in vitro cytotoxicity and apoptosis in cancer cell lines. Asian Pacific Journal of Cancer Prevention, 16(15), 6581-6589) and doi: 10.1016/j.mycmed.2015.11.004 (Haghdoost, N. S., Salehi, T. Z., Khosravi, A., & Sharifzadeh, A. (2016). Antifungal activity and influence of propolis against germ tube formation as a critical virulence attribute by clinical isolates of Candida albicans. Journal de mycologie medicale, 26(4), 298-305). With some modifications made for the maceration and extraction of our propolis sample.
Regarding “What concentration of methanol was used?”
99.8% methanol was used. In the line 457, the following text was added:
“…methanol (99.8%)…”
- Chapter 4.3 - Were the strains deposited in the public collection?
Reply:
No not yet. In the first place because the campus to which the authors belong does not have a collection of strains. Secondly, when we carried out the tests, the SARS-CoV-2 pandemic began shortly after and many of the institutions with strain collections had restricted access. We will do it as soon as the institutions allow it. However, the strains are available to the entire scientific community that wishes to use them.

Round 2
Reviewer 1 Report
The below suggestion has not been accomplished in the second version of the article. Perhaps, It was difficult to localize in the text. . It would be of great value to include an ATCC strain in the results if this is possible to compare them with the clinical isolates.Reviewer 3 Report
The manuscript was very well revised by the authors. The authors fully answered all questions of the reviewer. I recommend manuscript for publication.